# Point Prevalence of Complementary or Alternative Medicine Use among Children Attending a Tertiary Care Hospital

**DOI:** 10.3390/children10010132

**Published:** 2023-01-10

**Authors:** Angharad Vernon-Roberts, Abida Denny, Andrew S. Day

**Affiliations:** Department of Paediatrics, University of Otago, Christchurch 8011, New Zealand

**Keywords:** CAM, prevalence, disclosure, homeopathy, holistic, spiritual, Rongoā Māori, chronic, prescription, conventional medicine

## Abstract

Background: Complementary or alternative medicine (CAM) describes products/practices outside conventional medical care. CAM may be used to support or replace conventional/prescribed therapies. The aim of this study was to determine patterns of CAM use among children attending a tertiary care hospital in New Zealand (NZ) and measure parental opinion about CAM. Methods: Prospective survey-based study among children and their parents attending inpatient and outpatient clinical areas. Surveys collected demographic and health variables, current CAM use, and parental opinions on CAM. Results: Of the 236 children participating: 41% female, mean age 6.8 years (SD5), 76 (32%) with a chronic illness. CAM was used by 132 (56%) children, the most common being: oral supplements, body manipulation methods, or holistic practices. CAM use was associated with lower child health rating (*p* = 0.001), Māori ethnicity (*p* = 0.03), parent education level (*p* = 0.002), and family member CAM use (*p* < 0.001). Opinion survey results revealed CAM use was most strongly related to doctors recommending CAM, information on CAM, and CAM cost. There was a 31% CAM disclosure rate to the child’s medical team. Conclusions: This study highlights cultural differences in CAM use not previously reported among children in NZ. Parental opinion regarding CAM influences use for their child and disclosure rates.

## 1. Introduction

Complementary and alternative medicines (CAM) refer to products or practices considered to be outside of standard medical care. CAM are commonly used to support, or to replace, conventional allopathic treatments as well as to improve overall well-being [1,2]. The use of CAM is generally high among children [3], especially those with chronic health conditions who may utilize multiple CAM therapies [4,5]. Studies among general pediatric populations are less frequently reported, and little is known about CAM use for children with or without health conditions in New Zealand (NZ). One previous study reported CAM use among 70% of children attending NZ primary care practices and tertiary care hospitals, with 77% reporting that they had not disclosed their child’s CAM use to their medical team [6]. Non-disclosure of CAM is a common finding among pediatric populations [1,4,7,8,9,10] and while CAM have been used for years as part of traditional health maintenance, few have been rigorously tested for efficacy, reliability or safety [11,12,13,14]. Non-disclosure of CAM use to a child’s clinical team may increase the risk of adverse events such as interactions with prescribed medications, or poor compliance with prescribed medications [15,16,17].

Few public awareness resources are available for families in NZ regarding CAM, and CAM therapies are generally insufficiently regulated [18]. The results of this study may highlight the need to raise awareness for families, and improve physician communication regarding CAM, thereby reducing the chance of adverse events related to their use. In addition, measuring opinion and awareness of CAM therapies may highlight gaps or misconceptions that may be addressed. This study aimed to assess the frequency of CAM use among children attending a tertiary care hospital, and measure parent opinion of CAM in order to identify patterns of use and disclosure rates. 

## 2. Materials and Methods

### 2.1. Study Recruitment

Children and their parents were approached to participate in the research via face-to-face recruitment. Researchers [AVR, AD] attended clinical out-patient and inpatient areas at a NZ tertiary care hospital: Christchurch Hospital, Christchurch. 

### 2.2. Study Population

All children were included in the study and no limitations were set for age of the child, presence of a chronic illness, or any additional variables. Only parents or the primary care-giver could participate as they were considered to know most about CAM use for the child. Either the mother or father (or equivalent primary care-giver) could take part as this enabled analysis by gender of the reporter.

### 2.3. Outcome Measures

All outcome measures were completed by parents in the presence of the researcher using online forms on a tablet device provided by the research team. Surveys were developed from assessment tools available in the literature and adapted for the purpose of this study as no validated/standardized tool was identified that was fit for purpose.

#### 2.3.1. Demographic, Health and CAM Information

Socio-demographic and child health variables were collected relating to: age of the parent and child, gender of parent and child, parental education level, family ethnicity, postcode, the number of adults in the household, the number of children in the household, annual household income, prescription medications taken by the child, presence or absence of a chronic (underlying) health condition, the reason for the child to be attending hospital, the number of adults in the household using CAM, and the number of children in the household using CAM. The parents were also asked to rate their child’s overall level of health in the last week using a 0–10 scale. 

#### 2.3.2. Child CAM Use

Parents were provided with a guideline sheet to give comprehensive examples of CAM (Appendix A). They were then asked to complete a question on whether their child used CAM. If parents answered in the affirmative, they were directed to a survey asking a series of questions on each CAM used: name, source, source of information, frequency, cost, duration, reasons for use, disclosure to medical team, perceived benefit, and side effects. 

#### 2.3.3. Parent Opinion Survey

All parents, whether their child used CAM or not, were asked to complete an opinion survey regarding CAM use and acceptance. The survey consisted of 16 statements scored on a five-item Likert scale ranging from strongly disagree to strongly agree. For the purpose of presentation and analysis, results were subsequently condensed to ‘Agree’, ‘Neutral’ and ‘Disagree’. 

### 2.4. Ethics and Consent

All children were asked to provide written assent for their parents to provide information on them, and all parents provided written consent for participation. The study was granted ethics approval by the University of Otago Human Ethics Committee (Health) (H21/028) and Christchurch Hospital Locality Approval (RO21039).

### 2.5. Statistical Analysis

The dependent variable (CAM use) was analyzed against demographic and child health data, as well as parent opinion of CAM, to identify meaningful associations. Group comparisons between categorical and linear variables were performed by analysis of variance. The Chi squared (χ^2^) test of independence was performed to assess the relationship between ordinal variables, results presented as χ^2^ value and the Phi effect size, with results closer to 1.0 indicating a stronger effect size. Significance considered at a level < 0.05. Analysis performed using SPSS 27.0 [19]. 

## 3. Results

### 3.1. Participant Demographics

Two-hundred and fifty-eight children and their parents were approached to participate: 236 consented and completed the study, a response rate of 91.5% (Table 1). 

### 3.2. The Use of Complementary Alternative Medicines by Children

Of the 236 parents completing the survey, 132 (56%) utilized at least one CAM for their child, and 22 (9%) used more than one CAM, giving a total of 179 CAM used. Of those that did not use CAM for their child 57 (55%) said they would use CAM, 42 (40%) said maybe, and 5 (5%) said they would not use CAM. The most common type of CAM reported were oral supplements, followed by body manipulation practices, and holistic therapy, with 7% using Rongoā (traditional Māori remedies) (Figure 1). 

Parents were also asked details about each individual CAM used (Table 2). The main sources of CAM were from pharmacies (26%), online (20%), or CAM Specialists (19%). The main source of information regarding each CAM were family (37%) or friends (27%), with lower scores for Health Professionals (23%) or pharmacies (20%). The most frequent cost of the CAM the child was currently taking was $20–50 NZD (32%), with most taken daily (46%) and had been utilized for a duration of over 12 months (55%) (Table 2). The main reasons for using CAM for their child were for the treatment (42%) or prevention (34%) of symptoms, with only 22% using CAM for a chronic health condition. Few children experienced side effects (1%), and the majority saw improvement rated as ‘lots’ (40%) or ‘slight’ (46%). 

### 3.3. Factors Associated with CAM Use by Children

A number of variables were shown to have an association with the use of CAM among the study population (Table 3). A lower overall health rating within the previous week was strongly associated with increased CAM use (*p* = 0.001), although the child having a chronic illness (*p* = 0.096), or being on prescription medications (*p* = 0.05) did not affect CAM use. Identifying as being of Māori ethnicity (New Zealand indigenous population group) was associated with increased CAM use (*p* = 0.037), although the effect size was small, and this was not evident for the other represented ethnicity groups. Rongoā was used by 17 children in total: 10 (60%) with Māori heritage, 6 (35%) NZ European, and 1 (5%) Pasifika. Children were more likely to use CAM if the parent reporting was female (*p* = 0.029), and also if the parent had a higher education level (post-secondary education) (*p* = 0.002). The variables most strongly associated with CAM use for the child were whether the parents and siblings also used CAM (*p* < 0.001). 

### 3.4. Parental Opinion Survey

The results of the opinion survey showed that the majority of parents strongly agreed with statements regarding the need for interaction and support from their child’s doctor regarding CAM, although most did not require their doctor’s recommendation in order to use it (Table 4). Statements relating to safety, efficacy, side effects, and cost were scored as ‘neutral’ by the majority, and most parents thought that CAM therapists should be qualified and registered. The opinion survey results were studied for the relationship between each statement and CAM use for children (Table 4). Opinions relating to seven statements were shown to have significant associations with CAM use, with the strongest effect sizes being regarding only using CAM if their child’s doctor recommended it, and there being sufficient information available regarding CAM.

### 3.5. Associations with Non-Disclosure of CAM

Of all 236 parents reporting CAM use for their child, just 56 (31%) parents had disclosed this information to their child’s medical team. When all demographic and CAM related variables were tested for their association with disclosure only the child’s age [older children = increased disclosure] (*p* = 0.02, CI −3.1 to −0.3) and the category of CAM (χ^2^ 14.2, *p* = 0.015) were related (effect size 0.28). Individual analysis by CAM types showed that oral supplements were more likely to be disclosed (*p* < 0.001), and spiritual practices less likely to be disclosed (*p* = 0.023) (Table 5). There was no difference between disclosure rates for the different oral supplement types (χ^2^ 0.1, Phi 0.04, *p* = 1.0). Reasons for CAM non-disclosure were not collected as part of the CAM survey; however, a small number of parents relayed that the use of CAM may not have been relevant to disclose (such as prayer), and a few reported that were using therapies that were not recommended by their child’s doctor.

When testing for relationships between CAM disclosure and opinion statements, parental agreement with three were shown to have a negative relationship with parents disclosing CAM to the medical team:#6. Enough is known about the effectiveness of CAM (χ^2^ 9.3, Phi 0.2, *p* = 0.01)#7. Enough is known about the safety of CAM (χ^2^ 8.8, Phi 0.2, *p* = 0.01)#8. Enough is known about the side effects of CAM (χ^2^ 6.8, Phi 0.2, *p* = 0.03)

## 4. Discussion

This study presents patterns of CAM use among a population of children, with and without chronic health conditions, attending a tertiary care hospital in NZ. Associations with CAM use were shown for the child’s overall health, parent variables, and family CAM use, in addition to parent opinion. CAM disclosure rates were low but may be influenced by the type of CAM used.

The prevalence of CAM in the current study is marginally higher than that shown in a previous systematic review by Italia et al. [3] who reported a range between 8–48.5% point prevalence from 58 papers among the general pediatric population. However, that systematic review and other work has highlighted a problem of CAM reporting in that agreement on what is considered CAM differs considerably; therefore, impacting results and meaningful comparison with the wider literature [3,20,21,22]. While some of the CAM types included in the current study may not be considered CAM in comparative literature [21] (such as spiritual practices, nutritional supplements, traditional Rongoā Māori) formal guidelines have been published on what constitutes CAM [23]. This guideline includes all of those reported in the current study.

While the association between parent gender, education level, and family use of CAM in this study are common findings throughout the literature [3], associations with ethnicity are infrequently studied. In this current study, a relationship was shown between CAM use and those children having Māori heritage, with 72% of participants identifying as Māori using CAM, but there was no association among other ethnicity groups. The use of traditional Rongoā was used by 7% of the overall cohort, of which 60% were of Māori heritage, with Rongoā accounting for 83% of CAM use among Māori. The prevalence of Rongoā use is similar to a study carried out among NZ adults that also reported use at 7%, and although a higher proportion of users were of Māori heritage (83%), Rongoā only accounted for 11% of CAM use among Māori in the study [24]. The prevalence of CAM use among indigenous pediatric populations is infrequently reported, even when ethnicity data were collected [2,4,5,9]. In systematic reviews of indigenous adult populations, CAM use has been reported at 15–68% for those with diabetes [25], and 18–58% of those with cancer, with 58% prevalence among Māori [26]. The reasons for the high levels of Rongoā use in the current study may be due to increasing recognition of the importance of Māori health, with efforts to address health disparities becoming of national importance both in the clinical [27,28,29] and research settings [30]. This includes recognition of the importance of Rongoā and CAM in the healthcare setting [31,32,33]. These factors may be encouraging the use of Rongoā for children as it is becoming more widely understood and awareness among health care workers may be increasing.

The disclosure rates of CAM to the medical team in this study were low (31%), but comparable to previous studies among this population that have reported variable rates of 4–52% [5,10,11,34,35]. The anecdotal evidence provided by parents in the current study on reasons for non-disclosure is mirrored frequently in the literature, alongside other reasons such as: the information is not relevant [5,10,11,34,35], CAM is not harmful [5,10,11,35], their doctor does not agree with CAM use [11], the doctor did not ask [5,10,34,35] and concerns about getting a negative response from their doctor [5,10,34,35]. The frequency of parents reporting CAM use to their medical team was negatively associated with the specific use of spiritual practices in the current study, with associated anecdotal reports of ‘prayer’ not being important to disclose. However, while it may be considered that CAM such as spiritual practices (prayer and meditation in the current study) are relatively ‘hands-off’, the use of all types of CAM may have direct or indirect safety issues [36]. Indirect harm may be associated with CAM due to beliefs of efficacy causing a delay in the diagnosis or treatment of a medical condition, cessation of prescription drugs, social withdrawal, and extreme restrictive diets [22,36,37]. Direct clinical harm may be associated with the CAM therapy itself, for example by ingestion or topical application, interactions with conventional medications, incorrect diagnosis by CAM specialist, or body manipulation methods [22,36,37]. In the current study. the relationship between CAM disclosure and parental agreement with statements on there being sufficient information regarding CAM safety, efficacy, and side-effects, highlights this issue further and is concordant with literature stating that parents perceive CAM as being ‘safe’ [5,10,11,34,35]. In addition, with the main sources of information on the CAM used being reported as family, friends, and online sources, it may be surmised that there is insufficient information and education regarding CAM use and safety among the pediatric population that should be addressed. The clinical team treating children in any hospital or primary care setting should initiate conversations about CAM in an open, nonjudgmental way, and it should be incorporated into routine history and patient records for all patients at every visit [11,38]. With low levels of CAM knowledge reported among some doctors, nurses and pharmacists alike [18,39,40,41], detailed guidance on CAM, direct and indirect consequences, and patient communication regarding CAM should be provided in clinical areas where patient contact may occur.

### 4.1. Strengths

The study had a good representative sample size, and the indigenous Māori population of children were well represented, thereby allowing for cultural comparisons on CAM use. Recruitment took place in many clinical areas that allowed for associations to be made between those children with chronic illness, or taking prescription medications.

### 4.2. Limitations

The study would have benefited from collecting data on reasons for CAM non-disclosure by parents, as this would have allowed for a more in-depth analysis and critique against published literature. The assessment tool used for parental opinion was not validated, but no appropriate and validated tool was found that would not require adaptation, thereby invalidating the results.

## 5. Conclusions

This work has provided important information on CAM use among the pediatric population in NZ, and highlighted cultural differences that have not previously been studied. Parental opinions given in the study, as well as CAM disclosure rates, have raised a number of potential safety issues. While the findings are concordant with the literature, these issues should be addressed with improved education and awareness for parents, clinicians, multi-disciplinary team members, and pharmacists.

## Figures and Tables

**Figure 1 children-10-00132-f001:**
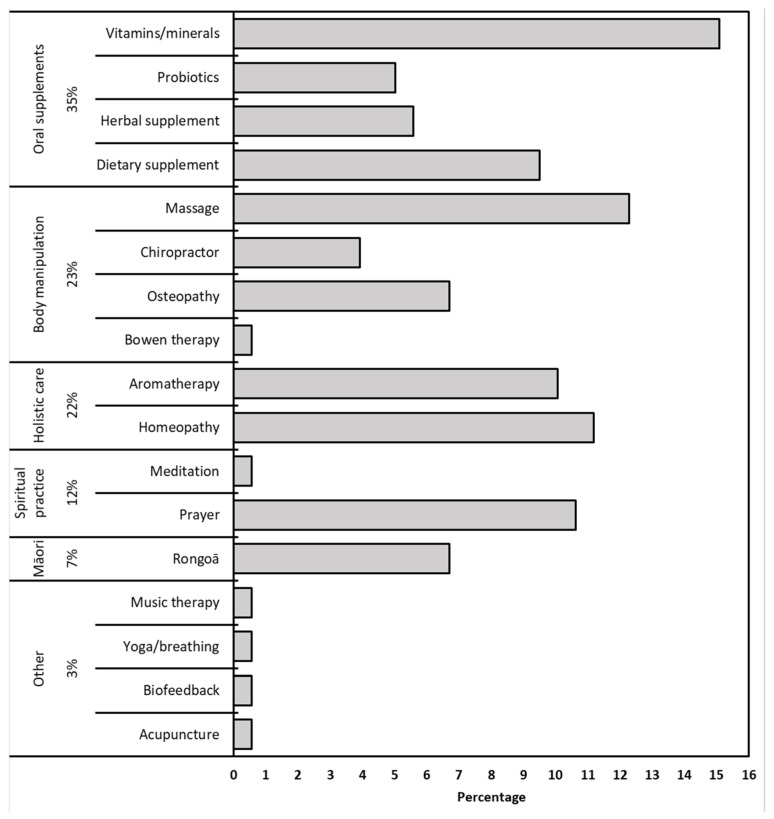
Frequency of CAM types used by NZ children.

**Table 1 children-10-00132-t001:** Demographic and health information of children and parents.

Variable	Category	All Mean (SD) or *N* (%)	CAM Used Mean (SD) or *N* (% *)	No CAM Mean (SD) or *N* (% *)
Child age	Years	6.8 (4.8)	6.9 (4.9)	6.9 (5.0)
Child gender	Female	97 (41)	62 (64)	35 (36)
Family ethnicity **	NZ European Māori Pasifika Asian MELA	203 (86) 46 (20) 13 (6) 9 (4) 4 (2)	110 (54) 33 (72) 5 (48) 7 (78) 2 (50)	93 (46) 13 (28) 8 (62) 2 (22) 2 (50)
Child chronic health condition	Yes No Don’t know	76 (32) 137 (58) 23 (10)	48 (63) 68 (50) 16 (70)	28 (37) 69 (50) 7 (30)
Child on prescription medications	Yes	86 (36)	54 (63)	32 (37)
Reason for attending hospital	Investigations Treatment: new diagnosis Treatment: existing diagnosis Accident/injury Other	48 (20) 69 (29) 58 (25) 39 (17) 22 (9)	31 (65) 37 (54) 30 (52) 23 (59) 11 (50)	17 (35) 32 (46) 28 (48) 16 (41) 11 (50)
Parent age	Years	37.4 (8.9)	37.7 (8.1)	37.0 (9.9)
Parent gender	Female	189 (80)	117 (62)	72 (38)
Parent education level	High School College University Post-graduate	92 (39) 43 (18) 72 (31) 29 (12)	40 (43) 30 (70) 42 (58) 20 (69)	52 (57) 13 (30) 30 (42) 9 (31)
Household income (NZD)	<$50 k $50–100 k $100–150 k $150–200 k >$200 k Not reported	36 (15) 67 (28) 49 (21) 30 (13) 17 (7) 37 (16)	22 (61) 35 (52) 30 (61) 13 (43) 9 (53) 23 (62)	14 (39) 32 (48) 19 (39) 17 (57) 8 (47) 14 (38)
CAM used by study children CAM used by parents CAM used by siblings	Yes Yes Yes	132 (56) 162 (69) 148 (63)	- 125 (77) 128 (86)	- 37 (23) 20 (14)
Child health in last week	1–10 scale (10 = best health)	7.2 (2.7)	6.7 (2.7)	7.8 (2.5)

MELA = Middle Eastern, Latin American, African, SD = standard deviation, N = number, % * = percentage of group, ** participants could choose more than one answer.

**Table 2 children-10-00132-t002:** Details relating to individual CAM use by children in NZ.

Variable	Category	Result N (%)
Source of CAM *	Pharmacy Online CAM Specialist Family member Supermarket CAM shop Friend Myself	47 (26) 35 (20) 34 (19) 30 (17) 28 (16) 26 (15) 19 (11) 18 (10)
Source of information on individual CAM *	Family member Friend Online Health professional Pharmacy CAM Specialist Book/magazine/newspaper CAM shop Church	67 (37) 49 (27) 48 (27) 42 (23) 36 (20) 18 (10) 18 (10) 15 (8) 10 (6)
Cost of current CAM	Free <$20 $20–50 $50–100 >$100	48 (27) 29 (16) 58 (32) 33 (18) 12 (7)
Frequency CAM taken	Daily Weekly Monthly When needed	83 (46) 41 (23) 8 (5) 46 (26)
Duration current CAM utilized	< 1 month 1–6 months 6–12 months >12 months Sporadic	16 (9) 35 (20) 25 (14) 99 (55) 4 (2)
Why use CAM *	Treatment of symptoms Prevention of symptoms Knowledge of it working for other people To complement conventional/prescription treatment Worry about side effects of conventional/prescription treatment More effective than conventional/prescription treatment Lack of confidence in conventional/prescription treatment Other	75 (42) 60 (34) 45 (25) 35 (20) 9 (5) 8 (4) 2 (1) 51 (28)
Side effects	Mild No	2 (1) 177 (99)
Benefit	Improved lots Improved slightly No change	72 (40) 82 (46) 25 (14)
Used for chronic condition	Yes	40 (22)

* participants could choose more than one answer.

**Table 3 children-10-00132-t003:** Association between demographic variables and CAM use by children.

Variable	Mean Diff	*p* Value
Child age	0.023	0.97
Parent age	−0.69	0.56
Overall health rating	1.1	*0.001*
**Variable**	**χ^2^ (Phi)**	** *p* **
Child gender (female)	4.0 (0.14)	0.05
Family ethnicity: NZ European	1.8 (0.09)	0.18
Family ethnicity: Māori	4.4 (0.16)	*0.037*
Family ethnicity: Pacific Peoples	1.7 (0.09)	0.19
Family ethnicity: Asian	1.8 (0.09)	0.18
Family ethnicity: MELA	0.06 (0.02)	0.81
Child chronic illness	4.7 (0.14)	0.10
Child prescription meds	3.9 (0.13)	0.05
Parent gender	4.8 (0.14)	*0.029*
Parent education	15.4 (0.26)	*0.002*
Parent CAM	137.5 (0.76)	*<0.001*
Siblings CAM	150.3 (1.0)	*<0.001*
Household income	6.4 (0.17)	0.27
Reason for attending hospital	2.1 (0.1)	0.71

Results in italics to highlight significant at level *p* < 0.05.

**Table 4 children-10-00132-t004:** Results of opinion survey and association between statements and CAM use by children.

Opinion Statement	CAM Use	Disagree N (%)	Neutral N (%)	Agree N (%)	χ^2^ (Phi)	*p* Value
1. Doctors should be supportive of people using CAM	No CAM Use CAM	5 (50) 5 (50)	28 (62) 17 (38)	71 (39) 110 (61)	7.9 (0.18)	*0.019*
2. Doctors should ask patients if they are using CAM	No CAM Use CAM	1 (50) 1 (50)	32 (49) 33 (51)	71 (42) 98 (58)	(0.07)	0.60
3. Doctors should know about CAM and be able to give advice	No CAM Use CAM	1 (20) 4 (80)	29 (50) 29 (50)	74 (43) 99 (57)	2.1 (0.1)	0.346
4. I would only use CAM for my child if a doctor recommended it	No CAM Use CAM	29 (30) 91 (70)	38 (62) 23 (38)	37 (67) 18 (33)	39.5 (0.41)	*<0.001*
5. CAM do not interfere with prescribed drugs	No CAM Use CAM	12 (27) 33 (73)	60 (48) 64 (52)	32 (48) 35 (52)	6.8 (0.17)	*0.033*
6. Enough is known about the effectiveness of CAM	No CAM Use CAM	35 (40) 52 (60)	49 (50) 49 (50)	20 (39) 31 (61)	2.4 (0.10)	0.30
7. Enough is known about the safety of CAM	No CAM Use CAM	27 (40) 41 (60)	51 (48) 56 (52)	26 (43) 35 (57)	1.1 (0.07)	0.566
8. Enough is known about the side effects of CAM	No CAM Use CAM	25 (36) 44 (64)	55 (49) 56 (51)	24 (43) 32 (57)	3.1 (0.12)	0.212
9. There is sufficient information available about CAM	No CAM Use CAM	22 (31) 50 (69)	56 (62) 34 (38)	26 (35) 48 (65)	19.8 (0.29)	*<0.001*
10. CAM have fewer side effects than prescribed or conventional treatment	No CAM Use CAM	10 (42) 12 (58)	65 (48) 56 (52)	29 (31) 64 (69)	10.9 (0.21)	*0.004*
11. CAM is more effective than prescribed or conventional treatment	No CAM Use CAM	27 (42) 38 (58)	64 (48) 70 (52)	13 (35) 24 (65)	2.1 (0.10)	0.349
12. CAM therapists/practitioners should be qualified and registered	No CAM Use CAM	7 (54) 6 (46)	29 (45) 36 (55)	68 (43) 90 (57)	0.6 (0.05)	0.748
13. CAM is used by people who lack confidence in conventional treatment	No CAM Use CAM	41 (40) 62 (60)	44 (49) 45 (51)	19 (43) 25 (57)	1.8 (0.09)	0.404
14. CAM is used by people due to a lack of conventional treatment for an illness or condition	No CAM Use CAM	31 (42) 43 (58)	55 (51) 53 (49)	18 (33) 36 (67)	4.7 (0.14)	0.094
15. CAM can be used to replace conventional treatment	No CAM Use CAM	38 (42) 53 (58)	48 (55) 40 (45)	18 (32) 39 (68)	7.7 (0.18)	*0.021*
16. The cost of CAM puts people off using it	No CAM Use CAM	10 (28) 26 (72)	58 (59) 40 (41)	36 (35) 66 (65)	16.1 (0.26)	*<0.001*

Results in italics to highlight significant at level *p* < 0.05.

**Table 5 children-10-00132-t005:** Disclosure rates and association with CAM types for children.

CAM Type	Disclosure Rate (N (%))	χ^2^ (Phi)	*p* Value
Oral supplements	29 (49)	11.3 (0.3)	*<0.001*
Body manipulation	11 (27)	0.8 (0.1)	0.39
Holistic care	13 (31)	0.1 ((0.02)	0.82
Spiritual practice	2 (10)	5.2 (0.2)	*0.023*
Rongoā	2 (17)	1.5 (0.01)	0.23
Other	1 (25)	0.1 (0.02)	0.75

Results in italics to highlight significant at level *p* < 0.05.

## Data Availability

Data are available upon reasonable request to the corresponding author AVR.

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
