# Peer review of "Point Prevalence of Complementary or Alternative Medicine Use among Children Attending a Tertiary Care Hospital"

_children, 2023, doi:10.3390/children10010132_

Round 1

Reviewer 1 Report

The paper deals with a relevant issue to public health (particularly children and Complementary and Alternative Medicine), it is well written and its sections are well elaborated and explained. There is a small set of questions which I feel can improve the paper, that support my recommendation that it should be accepted after minor revision.

In the section on materials and methods, it would be important to specify some aspects, such as: What were the criteria for the inclusion of children in the study (namely in terms of age)? How was the target population reached? How were contacts established? Was it irrelevant whether the father or mother of the child completed the survey? How were the questionnaires administered?

Author Response

Thank you for this valuable feedback. It was an oversight not to include this information in the original manuscript and all points have been addressed in the methodology section of the revised version.

Reviewer 2 Report

This paper contributes to our understanding of CAM use globally, including traditional Maori healing practices. The authors demonstrate that families in New Zealand use CAM at similar to slightly higher rates than in other study population, and disclose CAM use at somewhat lower rates. The analysis suffers from a lack of standardization in this field of inquiry, including validated instruments to identify CAM usage. However, the authors present their findings clearly and do not overgeneralize in their conclusions. 

There are 3 self-citations of author ASD but these appear to be appropriate.

Author Response

Unfortunately, very few validated tools are available to assess CAM usage in this specific population and we agree that this is important to highlight. This matter was addressed in the original manuscript discussion [Section 4.2. Limitations] but we have added additional information in the methods section of the revised manuscript [Section 2.3 Outcome Measures].